# Advances in Portable Optical Microscopy Using Cloud Technologies and Artificial Intelligence for Medical Applications

**DOI:** 10.3390/s24206682

**Published:** 2024-10-17

**Authors:** Alessandro Molani, Francesca Pennati, Samuele Ravazzani, Andrea Scarpellini, Federica Maria Storti, Gabriele Vegetali, Chiara Paganelli, Andrea Aliverti

**Affiliations:** Dipartimento di Elettronica, Informazione e Bioingegneria, Politecnico di Milano, 20133 Milan, Italy; alessandro.molani@polimi.it (A.M.); francesca.pennati@polimi.it (F.P.); samuele.ravazzani@mail.polimi.it (S.R.); andrea1.scarpellini@mail.polimi.it (A.S.); federicamaria.storti@mail.polimi.it (F.M.S.); gabriele.vegetali@mail.polimi.it (G.V.); chiara.paganelli@polimi.it (C.P.)

**Keywords:** portable microscopy, lens-less microscopy, photoacoustic microscopy, smartphone microscopy, Internet of Things, artificial intelligence, deep learning, cloud computing, point-of-care diagnostics

## Abstract

The need for faster and more accessible alternatives to laboratory microscopy is driving many innovations throughout the image and data acquisition chain in the biomedical field. Benchtop microscopes are bulky, lack communications capabilities, and require trained personnel for analysis. New technologies, such as compact 3D-printed devices integrated with the Internet of Things (IoT) for data sharing and cloud computing, as well as automated image processing using deep learning algorithms, can address these limitations and enhance the conventional imaging workflow. This review reports on recent advancements in microscope miniaturization, with a focus on emerging technologies such as photoacoustic microscopy and more established approaches like smartphone-based microscopy. The potential applications of IoT in microscopy are examined in detail. Furthermore, this review discusses the evolution of image processing in microscopy, transitioning from traditional to deep learning methods that facilitate image enhancement and data interpretation. Despite numerous advancements in the field, there is a noticeable lack of studies that holistically address the entire microscopy acquisition chain. This review aims to highlight the potential of IoT and artificial intelligence (AI) in combination with portable microscopy, emphasizing the importance of a comprehensive approach to the microscopy acquisition chain, from portability to image analysis.

## 1. Introduction

Microscopy, an optical-based system, has been instrumental in a variety of biomedical applications, encompassing histopathology, biochemical detection, and disease diagnosis [1,2,3,4,5]. Microscopes provide superior magnification and resolution capabilities that surpass human vision, facilitating the differentiation of tiny structures [6].

Microscopic investigations typically involve directing light from a source beneath the microscope onto the sample using a condenser [7]. The light, after traversing the specimen, undergoes diffraction [8] and is collected by the objective lens, resulting in a magnified image of the sample for observation through the eyepiece [7].

Optical microscopy includes a broad spectrum of techniques, including bright-field, dark-field, phase contrast, and fluorescence microscopy [9]. Moreover, microscopy extends beyond optical methods, incorporating electron [10], scanning probe [11], digital holographic [12], and photoacoustic [13] microscopy. Each of these methods offers unique advantages, dictated by their underlying physics and imaging properties.

Beyond healthcare, the applications of microscopy span diverse fields such as environmental monitoring, food quality analysis, forensic science, and mineralogy [1,14]. While affordable benchtop optical microscopes are increasingly available, there are also specialized high-end microscopes that present improved capabilities but come with higher costs and require skilled personnel for operation [15].

Recent advancements in technology and evolving industry trends have introduced complementary solutions that are transforming the conventional chain of image acquisition and interpretation in microscopy, as illustrated in Figure 1. These innovations prioritize compactness and portability for field applications [16,17], with a special emphasis on the use of smartphones to enhance accessibility for both professional and non-expert users [18,19,20]. While tabletop microscopes have the advantage of operating independently of Internet connectivity, relying instead on local connections to computers and digital cameras for image visualization and analysis, incorporated communication [21] and image processing [6] capabilities are trending solutions that are driving the development of portable microscopes. These systems integrate the Internet of Things (IoT) [17] to facilitate faster and more efficient data communication [22]. Coupled with cloud services [23] for data sharing and telemedicine [24] and deep learning (DL) algorithms for automated image processing and interpretation [25,26], these solutions offer new opportunities as well as new challenges for the field of microscopy.

Despite the numerous advancements in the field, there is a noticeable gap: a lack of comprehensive reviews that holistically address the entire microscopy acquisition chain. This paper aims to fill this gap by reviewing recent advancements in the microscopy acquisition chain and their impact on healthcare. It focuses on three main areas: portability, IoT integration, and the evolution of microscopy image analysis from traditional to deep learning approaches. The goal is to highlight the potential of a comprehensive approach to the microscopy acquisition chain, from portability to image analysis, and underscore the need for further work in this promising domain.

## 2. Portability

Portable microscopy has seen significant advancements in recent years, leading to the emergence of lightweight microscopes and Point-of-Care Test (PoCT) devices and techniques [27]. Traditional desktop devices, while powerful, are often bulky, expensive, and require professional expertise for operation. These factors confine sample analysis to specific clinical settings, limiting real-time detection and analysis, particularly in remote or resource-limited areas [17,28]. As a result, portable microscopy and PoCT devices have garnered considerable interest from researchers and engineers for on-site observation and telediagnosis [29,30].

The concept of PoCT microscopy was first explored between the 1930s and 1950s by McArthur, whose seminal works underscored the potential benefits of this technique [28,31,32]. With advancement in electronic and optical technologies, optoelectronic elements such as laser diodes, light-emitting diodes (LEDs), optical fibres, and binary optical components have become more compact and affordable. This has accelerated the development of portable biomedical testing platforms for microscopy [16]. A portable microscope setup typically features the following characteristics:Compact and lightweight design. The portability of these devices allows for use in diverse environments, extending their reach beyond traditional laboratory settings [16,17,33,34,35].Timely delivery of results. The ability to deliver nearly real-time results eliminates the need to wait for lengthy laboratory reports, facilitating quicker decision making [16].Accuracy. The qualitative and quantitative analysis provided by these devices needs to be comparable to benchtop devices and laboratory equipment to gain acceptance and adoption by the user community [16].Simple optical systems. Uncomplicated configurations ensure compactness, ease of use, replaceability of components, adaptability, and rapid prototyping [16,35].User-friendly operation. The device should require minimal training and be intuitive to use, making it accessible to a wider range of users [16].Low energy consumption. Minimal energy requirements allow operation for extended periods, which is crucial for field applications [35].Affordability. The setup must be cost-effective to encourage widespread application and accessibility [16].

The subsequent subsections delve into different approaches proposed for transitioning from tabletop to portable microscopes (Figure 2). Singlet and lens-less microscopy are discussed, along with techniques for super-resolution microscopy. We then focus on photoacoustic microscopy (p-PAM), which holds the potential to enable non-invasive, real-time imaging in a portable and cost-effective manner. Lastly, we explore smartphone-based microscopy, a prominent example of portable microscopy that has been extensively studied and holds great promise for the future [18].

### 2.1. Singlet Microscopy

Lens-based devices often face constraints due to the size, weight, and cost associated with their multiple lens components [36]. Despite the affordability of traditional microscope objective lenses due to industrial mass production, their assembly and alignment remain complex and costly [16]. In contrast, singlet microscopy employs a single lens element to focus light onto the image sensor or detector. This simplified configuration offers several advantages over traditional multi-lens systems, including compactness, affordability, ease of manufacture, and reduced optical aberrations [16,37]. These characteristics make singlet lenses particularly advantageous in the development of portable and cost-effective imaging devices.

However, singlet microscopy, while simple and attractive for portable applications, faces several challenges. One of the primary issues is chromatic aberration, a phenomenon where a lens fails to focus all colours to the same convergence point, resulting in a blurred or colour-fringed image. This is particularly problematic in singlet microscopy, as it uses a single lens element made of one material, making it difficult to balance the spectrum dispersion or chromatic aberrations. Moreover, maintaining high resolution when the distance between the bio-sample and image sensor increases necessitates the exploration of various lens designs, like graded index (GRIN) lenses, meta-lenses, and non-rotational symmetric lenses. However, each comes with its own set of challenges. While GRIN lenses effectively eliminate spherical aberrations, they struggle with off-axial aberrations at a large field of view (FOV) [38]. Meta-lenses, while thin and light, require time-consuming and expensive fabrications and pose challenges in surface measurements [39,40]. In non-rotational symmetric lenses, the net aberration is field-dependent, with different aberration components contributing uniquely to this behaviour, each varying in orientation and magnitude across the field [41].

To handle the challenges of the optical elements in singlet microscopy, some studies have proposed a combination of singlet lens microscopy and computational imaging, based on deep learning. For instance, Bian et al. [42] designed a singlet aspheric lens with high cut-off frequency and linear signal properties, coupled with a trained deep learning network, to transfer the monochromatic grayscale microscopy picture to a colourful microscopy picture, with only one single-shot recording by a monochromatic CMOS image sensor. Similarly, Shen et al. [37] designed a compact bright-field microscope, measuring approximately 10 cm × 10 cm × 20 cm and weighing around 400 g, that combines a singlet lens with a deep learning algorithm to enhance image resolution and contrast, improving the resolution from 2.19 μm to 1.38 μm. They further extended the application of singlet lenses to multi-spectral microscopy, applying deep learning algorithms to mitigate optical aberrations [43]. Additionally, Gordon et al. [44] developed a portable and low-cost fluorescent microscope for measuring malarial parasitaemia, using monochromatic illumination and a singlet aspheric objective lens to achieve sub-micron resolution with a long working distance.

### 2.2. Lens-Less Microscopy

Lens-less system design. Lens-less microscopy represents a further simplification in design compared to singlet microscopy, potentially leading to significant reductions in volume, weight, and cost. Despite these reductions, lens-less microscopy still enables high-resolution imaging of microscopic samples while offering a large FOV [45]. In this approach, the sample is placed between a charge-coupled device (CCD) or complementary metal-oxide-semiconductor (CMOS) image sensor and a coherent or partially coherent light source, such as lasers and LEDs. As the light source illuminates the sample, its diffraction pattern is directly captured by the sensor array. In contrast to lens-based imaging systems, which aim to project a potentially magnified replica of a scene onto an image sensor, the objective of a lens-less imaging system is to establish a reversible transfer function that links the incident light field with the sensor measurements. Computational algorithms are then typically applied to reconstruct the image [45,46,47,48,49,50,51]. The simplified concept of a lens-less microscopy setup is shown in Figure 3.

Lens-less techniques. There are three main approaches to lens-less imaging, specifically shadow imaging, lens-less fluorescence imaging, and digital holographic imaging, which have been widely reviewed in the literature [36,45].

Shadow or projection imaging is a simple form of portable lens-less microscopy [16,45]. The sensor-to-sample distance (z2) is typically smaller than the source-to-sample distance (z1) and less than 500 µm, to minimize diffraction effects [36]. The image sensor captures the grayscale shadow which appears either as blurry spots, in the case of incoherent light sources, or as concentric fringe interference patterns, with coherent light sources [45,52,53]. Shadow imaging does not rely on algorithms for image reconstruction; instead, it applies a pattern-matching approach to characterize and differentiate the recorded diffraction signals [45].

In lens-less fluorescence imaging, the excitation light determines an incoherent fluorescent emission from the sample with a different wavelength, which is captured by the image sensor [45,54,55]. Despite the cost-effectiveness, portability, and wider FOV advantages of lens-less fluorescence imaging devices, both hardware and computational solutions are required to improve resolution, which falls short of conventional benchtop microscopes [45,56].

Digital holographic lens-less microscopy computationally retrieves the complex optical field of intensity and phase information from a hologram. The hologram is the interference pattern generated by the superposition of the light passing through the sample and being scattered (object wave) and the reference wave passing without interaction [16,45,57,58]. Compared to shadow imaging, digital holographic reconstruction improves detection signal-to-noise ratio (SNR) and resolution [45,59]. In a holographic lens-less system, resolution is mainly limited by the pixel size of image sensors. Pixel super-resolution techniques are applied to overcome this limitation and to achieve nearly diffraction-limited imaging [45,60].

Challenges. Lens-less microscopy, while advantageous in terms of simplicity and cost, faces several challenges. These include maintaining image quality, managing data acquisition speed, and handling phase retrieval. Irregular illumination can introduce background noise, which can degrade the image quality. High-throughput imaging demands rapid data acquisition and processing, a requirement that can be particularly challenging in real-time applications. Additionally, inconsistencies in the focus plane across a large FOV necessitate further parameter calibration. Computational imaging methods have been employed to address these issues, achieving simultaneous wide-field and high-resolution imaging performance. A technique known as the “separation method” has been developed to manage noise. This method reconstructs both the illumination profile and the object, separating and eliminating the background noise from the object [61]. However, while digital signal processing can suppress noise, it can also obliterate some of the object features. This necessitates a careful balance between noise suppression and feature preservation, achieved by combining thoughtful system design, innovative noise suppression techniques, and computational methods.

Techniques for high-resolution imaging. Various techniques have been proposed for enhancing the resolution of lens-less on-chip microscopy, including the use of fibre-optic waveguides, synthetic aperture, wavelength scanning, and adaptive relaxed iterative phase retrieval. Bishara et al. [62] developed a lightweight lens-less on-chip microscope that uses fibre-optic waveguides and LEDs to capture super-resolved images of malaria parasites. They activated waveguides individually to obtain multiple holograms with subpixel shifts and then applied a pixel super-resolution algorithm to generate images with a resolution lower than 1 μm over a 24 mm^2^ FOV, surpassing the relatively large pixel size (~2.2 µm) of the sensor. Luo et al. [63] applied a synthetic aperture to lens-less on-chip microscopy and demonstrated its effectiveness on breast cancer tissue and Pap smears. They sequentially illuminated the sample at various angles using a partially coherent light source and then combined the captured images to reconstruct high-resolution images. In a separate study, Luo et al. [64] introduced a wavelength-scanning-based pixel super-resolution technique, as an alternative to physical shifting methods. They sequentially illuminated the specimen at different wavelengths within a narrow spectral range (480–513 nm), offering uniform resolution enhancement across the sensor array with fewer measurements. Wu et al. [65] applied wavelength scanning with a larger spectral range to achieve fast and motion-free quantitative phase imaging (QPI) of unstained live samples.

Zhang et al. [66] applied an adaptive relaxed iterative phase retrieval algorithm to achieve super-resolution reconstruction based solely on a stack of out-of-focus images, without requiring lateral displacements, wavelength changing, or illumination angle scanning. More recently, deep-learning-based approaches have emerged to reduce the computational burden of iterative methods and the need for multiple measurements [67,68].

### 2.3. Photoacoustic Microscopy

Physical principle. Photoacoustic microscopy (PAM) combines the high spatial resolution of optical microscopy with the deep tissue imaging capabilities of ultrasound (US) [69,70,71,72]. It works by converting light energy into ultrasound energy, which is then used to generate an image of the sample [13,73]. When a biological sample is illuminated with a pulsed laser, the absorbed light generates heat, causing the material to expand and produce acoustic waves that propagate through the tissue. These waves are detected by an ultrasound transducer (UST) and used to reconstruct an image of the sample. The amplitude of the acoustic waves is proportional to the absorption of light by the sample, allowing for mapping of light-absorbing structures. Endogenous molecules like haemoglobin can be visualized, providing label-free structural and functional information about blood vessels with high resolution. PAM can also be applied for molecular imaging, in which light-absorbing exogenous contrast agents are targeted [13,69].

PAM modalities. Acoustic-resolution photoacoustic microscopy (AR-PAM) and optical-resolution photoacoustic microscopy (OR-PAM) are two distinct modalities of photoacoustic microscopy, each with unique operational principles and imaging capabilities (Figure 4) [74]. AR-PAM operates using a weakly focused light beam as the excitation source. This results in a tighter acoustical focal spot size compared to optical focusing and allows for deeper penetration into biological tissues due to the lower scattering of acoustic waves compared to optical waves. However, this comes at the cost of a lower lateral resolution [75]. In terms of applications, AR-PAM is particularly useful for tasks that require imaging at greater depth, such as studying deep tissue structures or observing the progression of diseases that affect deeper layers of tissue. Conversely, OR-PAM employs a tightly focused light beam, with the focal spot of this light determining the lateral resolution. This results in a superior lateral resolution compared to AR-PAM, allowing for the visualization of fine structural details, such as capillaries and individual cells. However, the penetration depth in OR-PAM is limited due to the higher scattering of optical waves in biological tissue. This makes OR-PAM ideal for applications that require high-resolution imaging of surface or near-surface structures, such as dermatological studies or the examination of superficial tissue layers. In both AR-PAM and OR-PAM, the axial resolution is determined by the bandwidth of the ultrasonic transducer. Therefore, the choice between AR-PAM and OR-PAM depends on the specific requirements of the imaging task, such as the desired balance between resolution and penetration depth [74,76].

#### Portable PAM

Despite its potential for non-invasive, real-time imaging, progress towards portability for PAM has been slower compared to conventional optical microscopy systems [69]. To address this, researchers have focused on optimizing both the laser sources and the scanning system, using micro-electro-mechanical systems (MEMSs) and galvanometer scanning technology for light beam steering, for increased miniaturization and imaging speed [13,77,78,79,80,81,82,83,84,85,86,87,88,89,90,91,92].

Light source. The selection of the light source plays a relevant role in the design of portable PAM systems as it influences image quality, compactness, costs, and power consumption [89,93,94,95]. Compared to bulky and expensive solid-state laser sources used in traditional tabletop PAM setups, portable PAM systems use laser diodes (LDs) and light-emitting diodes (LEDs) as illumination sources, due to their compactness, cost-effectiveness, and lower power consumption. Specifically, LDs are preferred due to their higher power output, superior beam collimation, and greater spectral purity [78]. Indeed, low-cost portable LD-based OR-PAM systems have been developed, offering both high lateral resolution and compact design [96].

MEMS scanners. PAM scanning is achieved using micro-electro-mechanical systems (MEMSs) or galvanometer scanners. MEMS scanners, available in electromagnetic, electrostatic, and electrothermal types, offer precise and rapid scanning of laser beams [78].

Electromagnetic MEMS scanners have been widely used in PAM for their stability when operating in water [78,97,98]. For instance, Lin et al. [92] used a two-axis electromagnetic MEMS scanner in a hand-held OR-PAM device, achieving fast scanning and a volumetric rate of 2 Hz over an imaging domain of 2.5 mm × 2.5 mm × 0.5 mm.

Zhang et al. [82] integrated an electrostatic MEMS scanner into a hand-held photoacoustic imaging pen, providing a 2.4 mm FOV, a lateral resolution up to 18.2 μm, and an axial resolution of 137.4 μm.

Chen et al. [88] developed an ultra-compact OR-PAM probe using an electrothermal MEMS scanner. The 20 g probe, with dimensions of 22 mm × 30 mm × 13 mm, achieved a lateral resolution of 3.8 μm, an axial resolution of 104 μm, and an FOV of 2 mm × 2 mm and was used to image the vasculature of internal organs in rats and the human oral cavity. In another study [83], a wearable 8 g OR-PAM device was designed for imaging brain activities in freely moving rodents, achieving a lateral resolution of 2.25 μm. Dual-modality designs have also been proposed, integrating OR-PAM with other techniques such as optical coherence tomography (OCT) [87] and electroencephalography (EEG) [77].

Galvanometer scanners. Galvanometer scanners, which consist of mirrors mounted on galvanometers, reflect a laser beam across the sample in a raster [81,86,90] or rotatory pattern [80,89,99,100,101]. Compared to MEMSs, these scanners offer improved stability [13,78]. For instance, Hajireza et al. [90] designed a hand-held OR-PAM device measuring 4 cm × 6 cm and weighing around 500 g with a lateral resolution of ~7 µm that employs a 2D galvanometer scanning mirror system to steer the laser beam. In a different approach, Jin et al. [80] introduced a portable OR-PAM system that uses a rotary scanning mechanism. This design eliminates the relative motion between the imaging interface and the samples, providing a large FOV and a high spatiotemporal resolution. The system was successfully applied to the vascular networks in the human lip and tongue. Building on this, Qin et al. [99] further enhanced the system to achieve an even larger FOV of up to 40 mm in diameter laterally and 12 mm axially, while maintaining a high SNR.

### 2.4. Smartphone-Based Microscopy

Smartphone-based microscopy has garnered considerable research interest due to the advanced capabilities of modern smartphones. These devices, far more sophisticated than their “feature phones” counterparts, which are mobile phones with minimal characteristics and moderate pricing [102,103], offer powerful computing and connectivity features. However, for the purpose of this review, the terms “smartphone” and “mobile phone” are used interchangeably.

#### 2.4.1. System Design and Devices’ Classification

Smartphones, equipped with high-resolution cameras featuring CMOS image sensors, offer excellent image quality, small pixel size, high pixel count, sensitivity, and low power consumption. The potentiality of these devices is further expanded by miniature lens attachments that interface with the smartphone’s camera, providing the necessary magnification for microscopic observation [104]. Illumination, which is crucial for visualizing transparent or low-contrast samples, is typically provided by an integrated light source, often in the form of an LED. The sample under observation is placed on a specially designed holder or stage, allowing for precise positioning and focusing [15,105,106,107]. When combined with robust data processing circuits and substantial internal memory, smartphones transform into powerful imaging devices capable of visualizing micron and nanoscale particles [1,15,20,108,109,110,111]. The operation and image analysis of smartphone microscopes are facilitated by dedicated applications. These applications control camera settings, process captured images, and provide measurement and annotation tools. Moreover, these applications enable the sharing of images, fostering collaboration and remote diagnostics. Designed with portability and ease of use in mind, smartphone microscopes are compact, lightweight, and user-friendly. They offer a cost-effective and accessible solution for microscopic analysis, with potential applications ranging from healthcare diagnostics to educational purposes.

The classification of smartphone-based imaging devices depends on the functionality of the smartphone, the imaging technique realized, and the degree of system portability [30,105,112].

Functionality. In portable microscopy systems, smartphones serve as either detectors or interfaces [112]. As a detector, the smartphone’s built-in camera acquires images [108], with illumination provided either by the smartphone itself or external components like LEDs [105]. The images are then visualized through the pre-installed camera app on the mobile phone [113,114,115].

When used as an interface, the smartphone pairs with an external portable camera [116] or other image acquisition systems via a universal serial bus (USB) port [105], Bluetooth [117], or Wi-Fi connection [112]. In this configuration, the smartphone can be used to display the results on its screen [118], control the experimental setup [119,120], and process the captured images [121,122,123].

Imaging modality. The imaging modality of smartphone-based microscopes can be transmission, reflection, and phase-contrast mode [20]. Transmission is used to observe transparent samples like blood cells, while reflection is suited for imaging opaque and thicker objects. Phase contrast is suitable for observing transparent biological objects without the need for chemical dyeing or fluorescent labelling [124,125], although it requires more complex hardware [35]. Multi-illumination systems that utilize both transmission and reflection microscopy have also been introduced [122]. Additionally, smartphone-based fluorescence microscopy has emerged as a powerful tool. This modality leverages the sensitivity and specificity of fluorescence signals to provide detailed imaging of cellular structures and molecular interactions [126,127,128,129,130,131,132,133,134].

Portability. Based on their level of portability, smartphone-based microscopes can be categorized into hand-held devices and portable tabletop microscopes (Figure 5). Hand-held devices, typically characterized by a 3D-printed structure physically attached to the mobile phone [125,135,136,137,138,139], allow for easy transportation and use in various settings. On the other hand, portable tabletop microscopes include 3D-printed tabletop adapters [140] that exploit a fixing and focusing stage where the smartphone can be placed [104,110,141,142].

#### 2.4.2. Applications

Smartphone-based microscopes are attracting great attention due to their cost-effectiveness, portability, and fast analysis capabilities, especially for low-resource settings. Moreover, functions such as wireless communication, cloud storage, and GPS allow the real-time transmission of data to monitor and manage patients in central hosts. This technology has been applied in various fields, including blood sample analysis [15,115,143,144,145,146,147,148,149,150], detection of pathogens such as bacteria [126,151] and viruses [152,153,154,155], water and air quality assessment [1,118,119,122,137,156,157,158], and drug testing [159]. Beyond the biomedical domain, smartphone-based microscopes have found applications in the analysis of contaminants in food samples [160,161], education [162,163,164,165], and industrial settings [141]. In this section, we detail the biomedical applications of smartphone-based microscopy, including in haematology and infectious disease monitoring.

Haematology. Smartphone-based solutions offer a rapid, reagent-free, and portable alternative to traditional methods for blood cell analysis, which typically requires microscope visualization by trained experts or expensive flow cytometers. Zhu et al. [143] designed a cytometric system on a smartphone, achieving results comparable to benchtop haematology analysers. The system uses a phone camera attachment with 3D-printed replaceable components to measure white blood cells (WBCs), red blood cells (RBCs), and haemoglobin density. Their custom application processes the captured images to automatically count the cells, and the results are displayed on screen or transmitted to a server. Similarly, Bills et al. [147] proposed a cost-effective smartphone attachment incorporating a blue LED and an optical filter to automate WBC counting in a paper-based microfluidic system, showing similar performance to fluorescence microscopes while simplifying sample preparation. In another study, Janev et al. [148] integrated a paper-based filtration device with a smartphone fluorescence microscope for selectively staining and counting WBCs to screen urinary tract infections. The smartphone’s rear camera is coupled to a reversed lens module and a green bandpass filter to capture the fluorescence signal generated by a blue LED excitation. The smartphone-based microscopic system developed by Rabha et al. [15] achieved RBCs imaging with high resolution, high magnification, and a large FOV while requiring low computational power. The authors utilized a 3D-printed opto-mechanical system attached to the smartphone camera to house the optical components and the XY-stage and a cloud-based algorithm to post-process RBC images, minimizing the smartphone’s hardware requirements.

Image processing and deep learning have further enhanced smartphone-based microscopy in haematology. Mandal et al. [149] developed mSickel, an algorithm based on regional geometric properties for sickle cell detection, providing high accuracy and low computational complexity. A deep learning framework was implemented by de Haan et al. [115] to automatically screen sickle cells in blood smears with 98% accuracy. The images obtained with a 3D-printed unit attached to a commercial smartphone are enhanced by a first network to match laboratory microscope image quality. The second network differentiates healthy and sickle cells in the enhanced images by semantic segmentation. Pfeil et al. [150] adapted various instance segmentation deep learning models like region-based convolutional neural networks (R-CNNs) and YOLACT to detect and classify blood cell types from images captured by a low-cost mobile microscope, an ocular camera, and a smartphone. Additionally, Huang et al. [145] developed Smart-AM, a smartphone-based autofluorescence microscope that leverages a mask R-CNN algorithm to achieve over 90% accuracy in leukocyte subtype classification from blood smears. This system also enables label-free imaging by virtual staining of blood smears using a trained conditional generative adversarial network (cGAN). These works demonstrate how deep learning techniques integrated with affordable hardware can achieve high diagnostic accuracy.

Monitoring of infectious diseases. Smartphone-based microscopy has also been applied for the diagnosis of malaria and other parasitic infections. D’Ambrosio et al. [144] developed CellScope Loa, a mobile phone microscope with an automated stage for quantifying filarial parasites in blood. An iPhone 5s was used to control via Bluetooth connection an Arduino microcontroller that drives an LED array and a capillary sample holder moved by a servo motor to change the FOV, minimizing the intervention of healthcare workers.

For malaria diagnosis, several approaches have been explored. The smartphone LED and camera sensor were used by Stemple et al. [166] to illuminate and detect the light scattered from blood samples mixed with microbeads coated with antibodies specific to malaria antigens causing immunoagglutination. This approach showed high sensitivity to the target antigen with a detection limit of 1 pg/mL in 10% blood. Rosado et al. [167] applied supervised classification based on support vector machines (SVMs) to assess malaria parasites in blood smears. Notably, microscopic images were acquired exclusively with a smartphone equipped with an optical setup to achieve 1000× magnification. Yang et al. [168] and Nakasi et al. [169,170] both acquired thick blood smear images by mounting a smartphone onto the eyepiece of a benchtop microscope. They aimed to develop computationally efficient deep learning frameworks that can be run on smartphone applications. Yang et al. applied intensity-based Iterative Global Minimum Screening (IGMS) to screen parasite candidates and a customized CNN for classification, while Nakasi et al. presented an end-to-end deep learning pipeline for multi-class detection of malaria parasites and WBCs, employing pre-trained models and transfer learning. Similarly, Fuhad et al. [170] trained multiple classification models combining knowledge distillation, data augmentation, autoencoders, and feature extraction by CNN to ensure efficiency while maintaining high classification accuracy on 10 mobile phones with different computational capabilities.

Smartphone-based fluorescence microscopes in combination with assay techniques and microfluidic devices have also shown excellent performance for pathogen detection in bacterial and viral ranges. Müller et al. [126] tested a hand-held device using a laser diode and optical filters to image the signal from low concentrations of bacteria targeted with species-specific and general probes, demonstrating its potential in bacterial detection. Hutchison et al. [151] developed a method for detecting *Bacillus anthracis* spores using a smartphone microscope and a microfluidic incubation device, offering a simpler and cheaper solution than existing methods for detecting anthrax.

In the realm of viral infections, Wei et al. [153] demonstrated the first smartphone-based detection of single viruses. Using a 450 nm laser diode for excitation, a long-pass filter to reject scattered light, and an external lens for magnification, the authors were able to detect 100 nm fluorescent particles and individual human cytomegalovirus (HCMV) particles. Yeo et al. [152] further advanced the detection capabilities for H5N1 avian influenza virus with high sensitivity (96.55%) and specificity (98.55%). Chung et al. [154] detected norovirus, a highly infectious virus with a low infectious dose, in water samples, adding antibody-conjugated fluorescent particles and imaging the particles on a paper microfluidic chip with a smartphone microscope. Recently, Liang et al. [155] applied competitive particle immunoassays to detect SARS-CoV-2 antibodies in saliva, using smartphone microscopy for particle counting to distinguish between virus-negative and virus-positive samples in a simple and non-invasive manner [171].

In summary, smartphone-based microscopes are a versatile technology, with impactful applications in diverse fields. The advancements in image processing techniques, particularly deep learning, and integration with microfluidic devices have made them important alternatives to expensive laboratory equipment, especially in low-resource settings. Nonetheless, before smartphone microscopes can be widely used in clinical settings, several challenges remain, including rigorous clinical validation [172].

#### 2.4.3. Advantages of Smartphone-Based Microscopy

Smartphone-based microscopy offers significant advantages aligned with the benefits of portable devices. These include cost-effectiveness, compactness, low power consumption [173,174], and the ability to obtain real-time results [161,175]. The unique integration of smartphones enhances these benefits beyond those offered by conventional portable microscopes [176].

Cost-effectiveness and portability. Smartphone microscopes utilize the inherent hardware of existing smartphones, necessitating only a few additional components, often 3D-printed [105,116]. Their cost-effectiveness, coupled with their compact size and light weight, enhances their accessibility, especially in settings with limited resources [20,29,109,177].

Ease of use. Equipped with user-friendly interfaces and intuitive apps, these devices are designed for ease of use by non-experts. The intuitive apps guide users through the process of capturing and analysing images, making these devices versatile for both professional and educational settings.

Real-time analysis. Smartphone microscopes can capture and process images in real time, providing immediate results. This is particularly beneficial in medical diagnostics, where rapid results can enable timely treatment decisions.

Connectivity. The global coverage of mobile broadband networks, such as the Global System for Mobile communication (GSM) and the Long-Term Evolution (LTE), enables cloud-based services like cloud computing to improve image processing [15] and allows for real-time sharing of images and data as well, enabling information transmission to distant hospitals or medical centres, facilitating remote consultations and diagnostics [29]. Furthermore, smartphones can function as IoT devices, linking to a network of other devices and systems for more complex setups.

Standalone and autonomous devices. Smartphones serve as standalone and autonomous devices from the perspectives of image acquisition, computation, and communication. This autonomy is particularly beneficial in remote and resource-limited settings, where a single, portable, low-cost, and user-friendly device can expedite disease diagnosis and monitoring [178]. With their intrinsic telecommunication capabilities, smartphones can replace multiple diagnostic devices, enabling primary image analysis [179] to perform initial diagnoses and prevent the spread of diseases [180].

Integration with other smartphone features. Smartphones come equipped with a variety of features that can enhance the functionality of smartphone microscopes, such as GPS for geotagging images and accelerometers for measuring angles or movement during imaging, potentially enabling new imaging techniques.

Scalability. The capabilities of smartphone microscopes can scale with advancements in smartphone technology. Improvements in camera resolution, processing power, and software capabilities will directly enhance the performance of smartphone microscopes. Furthermore, as new applications for microscopic imaging emerge, smartphone microscopes can be adapted to meet these new demands, making them a highly versatile tool in the ever-evolving landscape of microscopy.

The number of smartphones has been steadily increasing worldwide since 2008 and is expected to continue to rise over the next five years. This trend underscores the potential reach and impact of smartphone-based microscopy in the years to come, making scientific investigation more accessible and user-friendly.

#### 2.4.4. Challenges and Developments

Smartphone microscopes, despite their portability and cost-effectiveness, face a range of challenges that span both hardware and software aspects.

Hardware. Smartphone microscopes encounter a variety of hardware challenges. A primary issue is the fixed magnification level, which restricts the detail level that can be observed [181,182]. Fixed-focus microscopic imaging, resulting from a non-adjustable working distance, is a significant challenge for smartphone microscopes [128,129,183]. Microscopic imaging requires precise control of the working distance between the microscope and sample, but it is often inconvenient to fix and precisely adjust the body of a smartphone to achieve this. Another notable challenge is device compatibility, as not all smartphone microscopes are compatible with different types of smartphones.

Modularity can address several of these challenges. A modular design allows users to easily swap between different sets of filters and lenses, thereby enabling the use of multiple magnification levels [182,184]. For tuning focus, an innovative solution has recently been proposed [185], based on the design of a seesaw-like structure capable of converting large displacements on one side into small displacements on the other, maintaining a constant displacement ratio. When an extended FOV is required, some arrangements proposed the use of an identical lens as the one present in the smartphone camera but reversed [29,164]. Moreover, a modular smartphone-based microscopic attachment can be used with different smartphones to address the compatibility issue.

Recently, learning-based image enhancement methods have been introduced to mitigate image degradation in smartphone microscopes, mapping smartphone microscope images to benchtop microscope images without requiring paired FOV data for training, thus further reducing device costs [185].

Software challenges. On the software side, the analytical performance of smartphone microscopes needs improvement. The automatic focus, exposure, and colour gain standard on mobile phones can degrade image resolution and reduce accuracy of colour capture if uncorrected [142]. Certain features of the on-board image processing software cannot be disabled, and phone-specific artifacts remain visible in mobile phone microscope images [142]. The accuracy and sensitivity of many smartphone-based tests must be improved to make them more reliable. Most tests detect only a single or a few targets. However, in clinical practice, there is often a need to test for multiple pathogens simultaneously because the clinical presentation is usually not entirely specific for a single infectious disease. Another significant software challenge is ensuring data privacy. Data transfer to a server or a cloud should be rapid and safe, and only considered when personal data privacy is guaranteed.

Balancing hardware and software solutions. Balancing these challenges involves a multi-faceted approach that includes collaborative development, iterative design, modular hardware design, development of advanced algorithms, standardization and regulation, and cost considerations. Cooperation with phone manufacturers and operating system developers is crucial to accelerate development of image-based diagnostic assays through access to the image processing pipeline [142].

There are several successful implementations and collaborative initiatives in this field. For instance, LudusScope [163] is an accessible, interactive do-it-yourself smartphone microscopy platform. Pocket MUSE is a compact smartphone microscope that demonstrates the first practical implementation of Microscopy with Ultraviolet Surface Excitation (MUSE) [128]. DIPLE, developed by SmartMicroOptics, is a product that can transform any smartphone or tablet into a powerful microscope. However, several significant hurdles such as clinical validation, standardization, security measures, and compliance with healthcare regulations/approvals need to be overcome to position mobile-phone-based imaging and sensing platforms at the front of POC applications [186,187].

## 3. Internet of Things

The Internet of Things (IoT) encompasses a vast network of physical devices that are embedded with sensors, software, and connectivity, allowing them to collect and share data. IoT enables these smart devices to communicate with each other and with other Internet-enabled systems, creating a vast network of interconnected units that can exchange data and perform various tasks autonomously [188,189]. As a result, automation, data collection, and improved efficiency are achieved in numerous domains of life. In the medical field, IoT is known as the Internet of Medical Things (IoMT) or Internet of Healthcare Things (IoHT). The Internet of Medical Things (IoMT) consists of a network of interconnected medical devices, wearables, sensors, and other health-related equipment, that securely exchange data with healthcare professionals. This exchange of information facilitates the development of innovative medical solutions and services, supporting remote monitoring, personalized healthcare, early detection of health issues, and improved patient outcomes [188,189,190,191].

The healthcare sector can undergo a significant transformation with the integration of IoT in the field of microscopy (Figure 6). By enhancing diagnostic capabilities, improving accessibility, and facilitating data-driven decision making, this technology holds immense potential. The advantages offered by IoT in this context are plentiful.

Remote Monitoring and Accessibility. The introduction of IoT-enabled microscopes has revolutionized the field of microscopy by allowing for remote monitoring and real-time observation of microscopy sessions from various locations worldwide. By harnessing the power of the IoT, these microscopes can instantly transmit real-time data, enabling the rapid transfer of high-quality images to distant locations. This capability has significantly expedited the diagnostic process, leading to more precise and efficient diagnoses, regardless of geographical distance [192]. This technological advancement has had a profound impact on scientific collaboration, as it enables scientists to offer guidance in telemedicine [193,194] and support remote project-based biology education in communities that lack sufficient resources [195]. Additionally, the integration of IoT technology on a global scale has enhanced screening programs and improved access to specialized healthcare, particularly in underserved remote areas with limited resources [196,197]. Furthermore, the integration of IoT has emerged as an effective solution to alleviate the strain on medical infrastructure and personnel [197,198].

Automation. The incorporation of IoT-enabled devices allows for the automatic modification of microscope settings, including focus, magnification, and illumination, depending on the specific sample being analysed. This leads to a decreased necessity for manual interventions, ultimately enhancing overall efficiency [199,200,201]. Furthermore, IoT facilitates the smooth integration of microscopes with different systems and devices [202]. Through automation, the dependence on manual adjustments is reduced, resulting in streamlined workflows and improved operational efficiency in healthcare settings where efficient time management is essential.

Enhanced Diagnostics and Treatment. IoT-enabled microscopy plays a crucial role in enhancing diagnostics and treatment by utilizing advanced image analysis algorithms and machine learning techniques, resulting in more precise and timely identification of subtle abnormalities and early disease indicators [198,203]. Such a framework is essential for the diagnosis and treatment of rare and chronic diseases, ultimately improving patient outcomes and enabling the development of personalized treatment strategies [5,193,204,205]. The efficient management of big data and time, facilitated by these devices, is essential in this context: the automation of processes and reduction of manual tasks through IoT technology enable the rapid delivery of services to individuals [200].

Integration with other medical systems and improved patient experience. The integration of IoT-enabled microscopy with electronic health records (EHRs) facilitates the seamless retrieval of images by medical professionals as needed. Additionally, the merging of IoT-enabled microscopy with laboratory information management systems (LIMSs) streamlines laboratory procedures, reduces errors in manual data entry, and enhances the efficiency of laboratory operations [206]. By optimizing workflows and reducing the need for invasive procedures or multiple appointments, IoT-enabled microscopy has the potential to improve the overall patient experience. Patients can benefit from quicker diagnoses, tailored treatment strategies, and reduced healthcare costs associated with unnecessary tests or interventions.

Cloud-based solutions. By combining the IoT with cloud solutions, it is possible to securely store and process data from IoT-enabled microscopes. Microscopy investigations often yield significant volumes of data, especially when employing advanced high-resolution imaging techniques. Cloud storage alternatives provide a convenient means of archiving microscopy data, facilitating an optimized process [207]. Moreover, cloud computing provides powerful computational resources that can be utilized for processing microscopy data [207]. By transferring computational tasks to the cloud, researchers can achieve faster processing times and perform more complex analyses compared to relying solely on local computing resources. A cloud-based microscopy simulation platform for electron microscopy, called cloudEMAPS and powered by cloud computing and modern server–client web programming architecture, has been described [208].

It is important to note that while this combination of technologies offers many benefits, it also presents challenges, particularly in terms of data security and privacy. Therefore, appropriate measures must be taken to ensure the secure transmission and storage of data, including the use of secure communication protocols, robust authentication mechanisms, and advanced encryption techniques [209]. Some solutions for ensuring data security and privacy include blockchain [198,210], Holochain [211], Fog computing [189,212], and Fuzzy-based Trust Management mechanism [190]. However, a further recurring problem is the use of legacy and outdated devices, which cannot be properly either updated or maintained, causing the risk of susceptibilities [213].

### Applications in Portable Microscopy

The advent of IoT has revolutionized the field of microscopy, opening a plethora of applications across various domains. Within the sphere of cancer research, microscopes empowered by the IoT have proven to be pivotal for quick diagnosis, especially in remote scenarios characterized by a shortage of medical professionals or the unavailability of cancer specialists. These advanced devices facilitate the classification of malignant cells and enable the prediction of cancer stages [2,198], bridging the gap in healthcare accessibility. These devices, equipped with advanced image processing algorithms, can analyse biopsy samples in real time, providing critical insights into the nature and progression of the disease [3]. IoT microscopes have facilitated the detailed examination of corneal images, aiding in the diagnosis and treatment of neuropathies and corneal diseases [193,214]. The power of IoT extends to the study of rare diseases as well, fostering collaboration between multiple hospitals [5]. By connecting microscopes in different locations, medical professionals can share and analyse microscopic data in real time, accelerating the pace of research and treatment development. Furthermore, in the field of biology education, IoT microscopes have transformed the learning experience. Students can now remotely access high-quality microscopic images, conduct experiments, and share their findings with peers and educators worldwide [194,215]. Thus, the integration of IoT in microscopy has not only enhanced scientific research and medical diagnostics but also enriched the educational landscape.

In this section, the application of IoT technologies to portable microscopy will be reviewed.

Ophthalmology. Remote screening for vision-impairing diseases is vital for early diagnosis and intervention, but traditional methods often face challenges in accessibility and timeliness, particularly in remote areas [216]. The use of IoMT and cloud technology can enhance remote patient monitoring and early symptom detection. Furthermore, big data analytics can integrate artificial intelligence and machine learning into healthcare systems.

Technological advances have enabled the transformation of smartphones into ophthalmic fundus cameras through attachments and lens adaptors [217,218,219]. Despite the benefits, many researchers recommend maintaining traditional imaging techniques due to their high sensitivity. Managing large-scale data transmission through smartphones without a dedicated IoT platform can be challenging [220,221].

Both Kavitha et al. [222] and Das et al. [223] have proposed scalable, cloud-based teleophthalmology frameworks using IoMT for detecting and predicting the progression of age-related macular degeneration. These systems use wearable head-mounted cameras and deep learning applications for disease severity detection. Retinal images, once captured, are encrypted and securely transmitted to cloud storage. This allows for personalized detection of disease severity and prediction of disease progression. Kavitha et al.’s system [222] is further aided by an optimal generative adversarial network (OGAN) for image analysis and severity computation, showing improved accuracy. Both architectures include a cloud computing framework for managing connections and data and an ophthalmologist dashboard for secure access to patient data, ensuring data privacy.

IoMT frameworks have been developed for diagnosing diabetic retinopathy [203,224]. These frameworks use blood glucose sensors for continuous monitoring and categorize the data into Type I or Type II diabetes. Depending on the diagnosis, patients are guided to consult a doctor or capture eye fundus images. A dedicated iPhone application in conjunction with a smartphone fundus camera is used in [203], whereas an IoT-enabled head-mounted camera is used in [224]. Both systems provide a cloud-based diagnosis outcome and (based on the diagnosis) advise patients to consult with ophthalmologists for further treatment.

An innovative IoMT platform [214] has been developed for remote eye examinations and early detection of corneal diseases. This platform integrates a lightweight, portable optical instrument with a cloud-based system. The instrument, equipped with adjustable focal distance and camera positioning, captures high-quality images of the corneal surface. The cloud-based system features an ophthalmologist dashboard, equipped with dual camera entries, camera selection, a scrolling system, and a photo/video capture button, simplifying remote eye examinations. Additionally, the platform guarantees swift transmission of the patient’s corneal images.

Oncology. While the application of IoT and deep learning models in the field of oncology has been a major area of interest in recent research [198,225], it is important to note that only a limited number of studies are currently available in the literature specifically focusing on the development of an IoT framework in portable microscopy. This highlights the emerging nature of this research area and the potential for further exploration and development.

Sampaio et al. [196] developed a framework that uses the µSmartScope device and IoT technology to capture microscopic images from liquid-based cytology samples for the automated detection and classification of cervical lesions by deep learning. Similarly, Bibi et al. [4] and Karar et al. [198] proposed IoMT-based frameworks for the automatic identification and classification of leukaemia subtypes. These frameworks use an IoT-enabled microscope to collect blood smear images, which are then uploaded to a leukaemia cloud for diagnosis using deep learning models, including an advanced AC-GAN model [198]. The diagnostic results are displayed on the clinician’s computer, facilitating appropriate medical care for leukaemia patients. In another study, Skandarajah et al. [226] reported the feasibility of a telemedicine-based oral cancer screening method using an automated tablet-based mobile microscope, known as the CellScope device. This method combines a minimally invasive technique called brush biopsy with a simplified staining protocol to capture high-resolution images, which are remotely evaluated by clinicians. The clinical utility of this framework was further explored by integrating an ANN-based risk stratification model [227].

In the field of dermoscopy, a multitude of IoMT devices have been developed for the classification of skin lesions [228,229]. However, the practical application of these devices in a home setting remains under-explained. IoMT-assisted frameworks [230,231] have been proposed for remote data collection and processing. These frameworks employ secure data transmission models for cloud storage and data-driven methodologies developed using deep learning models to classify skin lesions. The outcomes play a pivotal role in guiding patient care, encompassing self-care, medication, and scheduling doctor appointments. The corresponding mobile application, accessible to healthcare professionals globally, facilitates preliminary analysis and classification. This technological advancement significantly benefits rural health centres by enabling early detection of potential lesions, thus reducing treatment costs and promoting awareness about skin cancers.

Fingernail analysis. Lee et al. [232] introduced a novel framework for portable microscopy, leveraging an UPMOST (UPG622) DMC microscope camera sensor and an Android smartphone to integrate into a nail analysis capture device through the USB transmission interface. This framework incorporates a blockchain structure to ensure privacy, data integrity, trust, and traceability of modifications. The smartphone pre-processes the images and forwards them to data management centres. These centres are interconnected to establish a fingernail network, facilitating the function of nail recognition. For nail biometric authentication, the study employs the Histogram of Oriented Gradients (HOG) and the Local Binary Pattern (LBP) and compares the accuracy of various machine learning algorithms.

Smart Fully Integrated Lab. While smartphones have been primarily used as analytical systems in healthcare technology, their potential extends to developing fully integrated smart devices capable of executing comprehensive diagnostic procedures. In this context, a smartphone-based analytical/diagnostic device (SAD) was designed by Golmohammadi et al. [233]. This compact device integrates various components including a stirrer, centrifuge, microscope, and an analyser equipped with an optical system. It also includes a mobile app for image processing and an IoT/Fog-based model. The system was tested for early proof-of-concept diagnosis and continuous therapy monitoring of phenylketonuria, as well as for tuberculosis diagnosis, yielding promising results.

General Remote Purposes. Several studies in the literature have expanded the application of IoT microscopes beyond specific medical frameworks, recognizing their potential benefits for both diagnosis and research. For instance, a smartphone-based microscope has been proposed that can implement both bright-field and fluorescence microscopy [24]. This microscope, which utilizes an OPPO Reno 10X mobile phone equipped with a telephoto lens module and an infinite correction objective, can function as an IoT sensing node due to its inherent wireless communication capabilities, further enhanced by 5G technology. The device was used to scan stained hippocampal slices, and the images were analysed using the Surf algorithm for feature point detection and subsequently stitched together using image mosaic algorithms, yielding excellent results.

Another innovative solution is a portable digital microscope designed to capture the microbial structure in a Petri dish and transmit the images directly to computers via Wi-Fi [200]. This compact, portable system offers auto-focusing and remote-control capabilities. It employs Arduino as the microcontroller to facilitate interaction between various inputs and outputs, leading to significant time savings in microbial analysis.

Additionally, a 3D-printed portable robotic mobile microscope has been developed [110]. Despite its lightweight and low-cost design, it is equipped with a Bluetooth module for local communication, a Wi-Fi and broadband cellular modem for remote connections, and an Amazon Simple Storage Service (Amazon S3) bucket for image storage. The system ensures security through the industry-standard AES-256 encryption algorithm and Secure HTTPS through Transport Layer Security. This device has demonstrated effective diagnosis results for histopathology and infectious diseases such as soil-transmitted helminthiasis.

## 4. Microscopic Image Processing: Requirements for Portable Devices

Mobile microscopes, leveraging consumer electronics like smartphones and tablets, face challenges in matching the image quality of high-end benchtop microscopes used in medical diagnostics and clinical applications, which benefit from optimized illumination and optical systems. Portable microscopes, in contrast, need to produce high-quality images while ensuring affordability, compactness, and lightweight design.

These challenges stem from the physical structure and mechanical parts, often produced using fast prototyping methods like 3D printing, leading to misalignments in the microscopic setup regarding optical, mechanical, and illumination components and causing image distortions such as chromatic aberrations. The resolution is often limited due to the low numerical aperture and high aberration coefficients of the lenses in their simplified imaging systems. Even with advancements in mobile-phone camera lenses, large-scale fabrication techniques can cause random deviations in each unit, influenced by factors such as battery status and user experience. Additionally, most optoelectronic imagers in consumer electronics are designed for close and midrange photography, not microscopy, leading to additional distortions. The small pixel sizes of mobile-phone cameras needed to improve spatial resolution further limit their light sensitivity.

Figure 7 provides an overview of the general image processing pipeline for portable microscopes addressing these challenges. Image enhancement, segmentation, and classification algorithms can be executed locally or in the cloud to improve the output image quality and provide rapid automated analysis.

### 4.1. Image Enhancement

To improve image quality in portable microscopy, specific algorithms are employed to reduce noise caused by weak light and inexpensive filters, thereby improving the signal-to-noise ratio (SNR). Notable contributions in this area include the work of Kuhnemund et al. [234], who engineered a cost-effective mobile-phone-based microscope for on-site diagnostics using a random forest approach to differentiate noise background from real signal, and Zhao et al. [235], who utilized a content-aware image restoration (CARE) network to denoise images captured with a high-speed portable reflectance confocal microscope.

The limited depth of field, a characteristic inherent to optical microscopy, can cause some structures to appear unfocused. Extended Depth of Field (EDoF) algorithms are employed to merge a stack of slides, each taken at different focal positions along the optical axis, into a single, entirely focused composite image. However, the effectiveness of these algorithms depends on consistent image alignment and magnification, which are often compromised in portable devices due to the inferior quality of their mechanical components. Albuquerque et al. [236] introduced a pre-processing workflow for low-cost microscopy devices, investigating several pre-processing methods and two deep learning models based on convolutional neural networks (EDoF-CNN-Fast and EDoF-CNN-Pairwise) to generate EDoF images.

General image processing algorithms can be used for image restoration and enhancement. However, these models are hard to estimate theoretically or numerically and are effective in specific hardware settings and operation environments. Rivenson et al. [237] pioneered a universal algorithm for enhancing images and correcting distortions introduced in a mobile-phone-based microscope using a deep convolutional neural network trained using a supervised learning approach with smartphone microscope images as input and corresponding benchtop microscope images as labels. Once trained, the network enhances the input images in terms of spatial resolution, signal-to-noise ratio, and colour response, aiming to match the quality and FOV of a high-end benchtop microscope. Similarly, Haan et al. [115] proposed a neural network approach based on U-net architecture to standardize images and improve their quality in terms of spatial and spectral features. These variations stem from changing exposure time, aberrations, chromatic aberrations due to source intensity instability, mechanical shifts, and more.

### 4.2. Segmentation and Classification

The field of microscopic image segmentation and classification has recently seen a significant shift from conventional algorithms to machine learning (ML) and deep learning (DL) strategies [238,239,240,241]. While traditional algorithms can achieve desired accuracy, they are susceptible to errors due to minor changes in image acquisition parameters such as contrast, exposure, and resolution [238]. This has limited their use on mobile devices. In contrast, ML and DL strategies offer robustness against these changes, providing higher accuracy, flexible deployment, and superior generalization ability [238].

These strategies are being increasingly used in current research in blood cell analysis, particularly focusing on networks for semantic or instance segmentation of microscopic images [242,243,244,245]. In this context, various deep learning methods have been investigated, such as, for instance, segmentation of all blood cell types from images acquired with a low-cost mobile microscope [150], achieving the detection of 93% of ground truth blood cells.

Several studies have demonstrated the automated identification of parasites using images captured solely with affordable and readily available smartphone microscopes [167,246]. These studies leveraged machine learning techniques, utilizing geometric, colour, and texture characteristics of the images for accurate parasite detection. Deep learning algorithms have been employed in a wide range of diagnostic applications of portable microscopes. For instance, semantic segmentation of blood smears has been used in the diagnosis of sickle cell disease [115] to distinguish between normal RBCs, sickle cells, and the background. In another application, deep learning models were explored to identify and classify cervical cells from conventional cytology images, aiding in the detection of cervical lesions [196,247]. Furthermore, the composition of kidney stones has been assessed using smartphone-based microscopic images of surgically extracted stones [248].

Machine learning has also been applied to mobile holographic microscopy for air quality monitoring [118]. The algorithm was trained on size-calibrated particles to map detected spatial characteristics to particle diameters, helping to avoid detection errors and over-counting. The algorithm used spatial features from holographic particle images to develop a regression model that maps these features to particle sizes. The model was trained using manually sized particle images from a standard benchtop microscope.

These applications highlight the potential of ML and DL strategies in microscopic image segmentation and analysis. However, the acquisition of suitable datasets for training remains a challenge across these applications. The accuracy and generalizability of deep learning approaches are often hindered by the insufficient data problem that results from the high expense of human and material resources for microscopic image acquisition and annotation. Various strategies for data augmentation have been explored to overcome the limitations of the mobile-acquired image dataset specifically collected and manually annotated by specialists, such as synthetic data generation with generative adversarial networks (GANs) [249] and transfer learning [196].

Moreover, implementing these algorithms on mobile devices can be challenging due to limitations in processing power and memory. Deep learning architectures, specifically designed for mobile applications, have been explored in the realm of portable microscopy, such as MobileNet [196] or EfficientNet [250]. These architectures are optimized for the constrained computational resources of mobile devices, striking a balance between the model size, computational demands, and performance efficacy. In this context, cloud computing can significantly enhance the capabilities of portable microscopic devices, providing virtually unlimited storage and processing power, which can be particularly useful for handling large volumes of high-resolution microscopic images. With cloud computing, the training process can be offloaded to the cloud, and once the model is trained, inference can also be performed on the cloud, reducing the computational load on the device. However, using cloud computing for microscopic image analysis on portable devices also presents some challenges, including ensuring the privacy and security of patient data, the dependence on the quality of the network connection, and the cost of cloud services, which can be a concern, especially for resource-constrained settings.

## 5. Discussion

The traditional approach to microscopy is being reshaped across the entire acquisition and interpretation chain. This transformation is driven by the increasing need to find cheaper and smarter alternatives to very accurate but expensive technologies, in an attempt to democratize scientific research and healthcare. The transition from conventional tabletop microscopes confined to laboratories to portable microscopy systems that can be used directly in the field required numerous technological efforts for both hardware and software optimization.

Laboratory microscopes are typically bulky due to the need for sophisticated and precise optics that ensure the highest image quality and avoid distortions and aberrations. Compacting these systems involved reducing the optical components used, ultimately leading to the development of lens-less systems, while also focusing on emerging techniques like portable photoacoustic microscopy (PAM). The possibility of 3D printing the setups helped to cut the costs. Additionally, the use of smartphones, which have now become globally available tools, has been a major innovation. Thanks to their increasingly advanced optics, camera sensors, and computational power, smartphones have become powerful standalone tools for microscopy. Their widespread availability guarantees simplified and user-friendly experiences for all users.

The high connectivity and communication capabilities of smartphones and smart microscopes connected to the Internet allow for the exploitation of IoT towards IoMT. This facilitates data sharing and transmission from isolated locations, as well as storage and processing in the cloud, enabling remote evaluation and diagnosis.

The use of simplified optics and the resulting lowered image quality, the generation of large amounts of data, and the need to run analysis on more compact systems with limited computing resources make it necessary to integrate these compact microscopy systems with increasingly effective but lightweight algorithms. To this end, deep learning offers notable advantages for matching the image quality of laboratory systems and for automatically and quickly extracting relevant information.

These innovations along the microscopy workflow simplify and improve the overall process. Non-professional personnel can easily acquire images remotely, which can be sent via the Internet to cloud services where they are processed by deep learning algorithms and confirmed by healthcare personnel in central hospitals. This empowers patients and lowers the pressure on the hospitals workers in both developed and developing countries.

However, an important aspect that cannot be overlooked is specimen preparation. In medical and biological investigations, proper specimen handling is a critical prerequisite for accurate imaging and diagnosis. Preparation techniques vary widely from sample to sample, with specific protocols and tools. While innovations in portable microscopy focus on simplifying image acquisition, visualization, analysis, and interpretation, these instruments still require the sample to be manually prepared by experts, and the accuracy of the results depends on the correctness of the procedure [251]. This factor limits the ease of use in field settings and real-world applications, challenging the widespread adoption of portable microscopy. Despite some efforts to automate specimen preparation with microfluidic and optofluidic devices [251,252] or to simplify the sample preparation procedure for field acquisitions [253], this issue remains under-addressed in research on portable microscopy solutions.

Several other challenges remain. One of these is data security and privacy. With the transmission and storage of sensitive medical data in the cloud, robust encryption and secure communication protocols are required. Additionally, relying on Internet connection and cloud computing introduces issues related to network dependency with potential costs that can be prohibitive in resource-constrained settings. In such environments, this dependency potentially compromises the overall effectiveness of these systems by making cloud-based artificial intelligence and data transmission inaccessible. In contrast, traditional desktop microscopes can function entirely offline, and with their simpler design, they are often more robust and reliable, making them still the preferred solution where network access is poor or absent.

Improving machine learning and deep learning algorithms for image analysis is needed, not only in terms of accuracy but also in developing lightweight models that can be deployed on devices with limited hardware. Portable microscopes should be able to perform local computations and store the results temporarily for later transmission. Moreover, clinical validation is mandatory to ensure the accuracy of the system’s diagnostic capabilities, while continued research and development for cost-effective solutions is crucial to make this technology accessible to a wider audience.

## 6. Conclusions

In conclusion, the integration of the Internet of Things (IoT) and artificial intelligence (AI) in portable microscopy represents a significant advancement for medical diagnostics. Although several challenges still need to be overcome, the potential benefits are valuable in terms of improved accessibility, efficiency, and accuracy.

The future of portable microscopy lies in further refining the integration of IoT and AI. While we have highlighted how each of these areas individually has shown promising advancements, relatively few studies have specifically explored the entire microscopy investigation chain incorporating all three elements: portability, IoT, and deep learning. Continuing development in this direction is a promising research field, aiming to enhance POC diagnostics and improve healthcare in resource-limited regions.

## Figures and Tables

**Figure 1 sensors-24-06682-f001:**
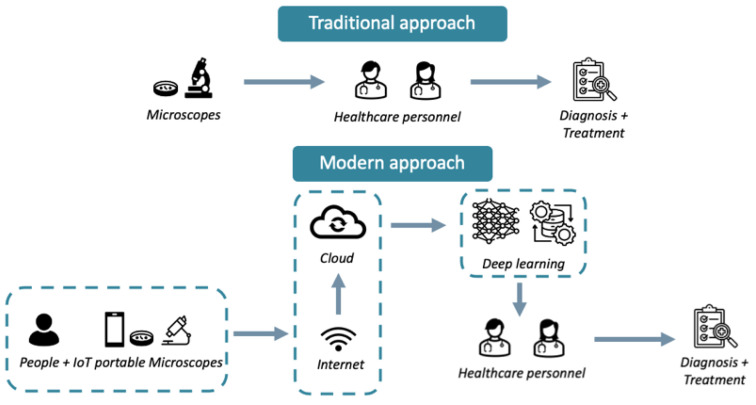
Schematic representation comparing traditional and portable microscopy approaches for image acquisition and analysis. The top panel illustrates the traditional method, involving laboratory-based microscopes used by trained personnel for clinical decisions. The bottom panel depicts the emerging approach based on compact microscopes operated by common people, with data transmitted over the Internet and processed by deep learning algorithms to facilitate the clinical decisions by healthcare personnel.

**Figure 2 sensors-24-06682-f002:**
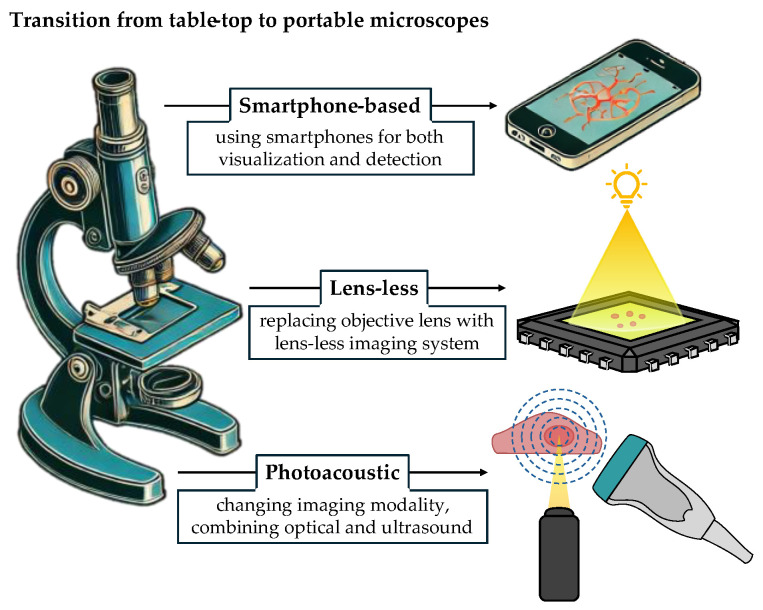
Recent solutions and technological advancements for transitioning from benchtop to compact portable microscopy. Smartphones can replace conventional visualization systems and serve as detectors. Solutions with a reduced number of lenses or lens-less can match the imaging capabilities of traditional lens-based microscopes. Photoacoustic microscopy (PAM) represents an alternative microscopic imaging modality, combining optical and ultrasound techniques.

**Figure 3 sensors-24-06682-f003:**
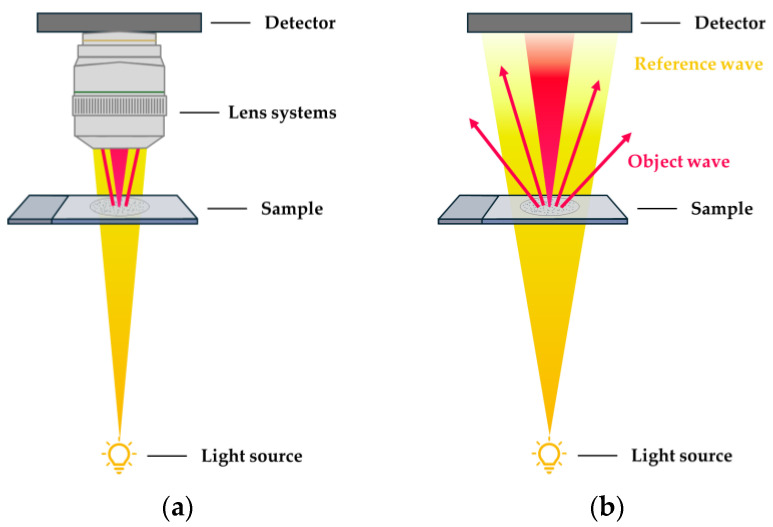
Simplified schematics of lens-based and lens-less microscopy. (**a**) In traditional lens-based microscopy, the detector captures an image of the sample magnified by the lens system; (**b**) lens-less microscopy eliminates the use of lenses, and a detector directly captures the light scattered by the sample.

**Figure 4 sensors-24-06682-f004:**
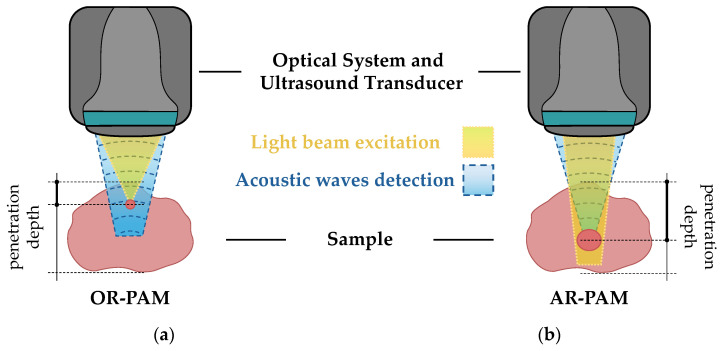
Simplified representation of photoacoustic microscopy principle and comparison between optical-resolution (OR-PAM) and acoustic-resolution (AR-PAM) photoacoustic microscopy. (**a**) In OR-PAM, a tightly focused light beam is used, resulting in high lateral resolution but limited penetration depth; (**b**) in AR-PAM, the acoustic focus is sharper than the optical focus, providing greater penetration depths but a lower lateral resolution.

**Figure 5 sensors-24-06682-f005:**
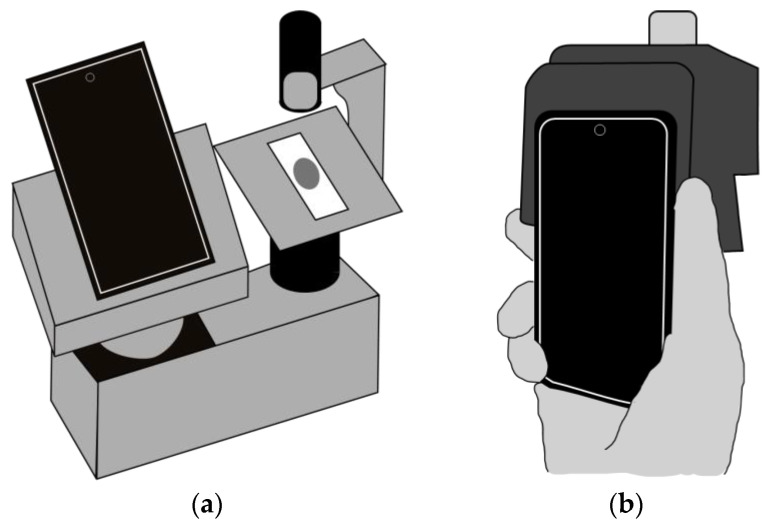
Examples of smartphone-based microscopes. (**a**) Portable tabletop microscope; (**b**) hand-held microscope.

**Figure 6 sensors-24-06682-f006:**
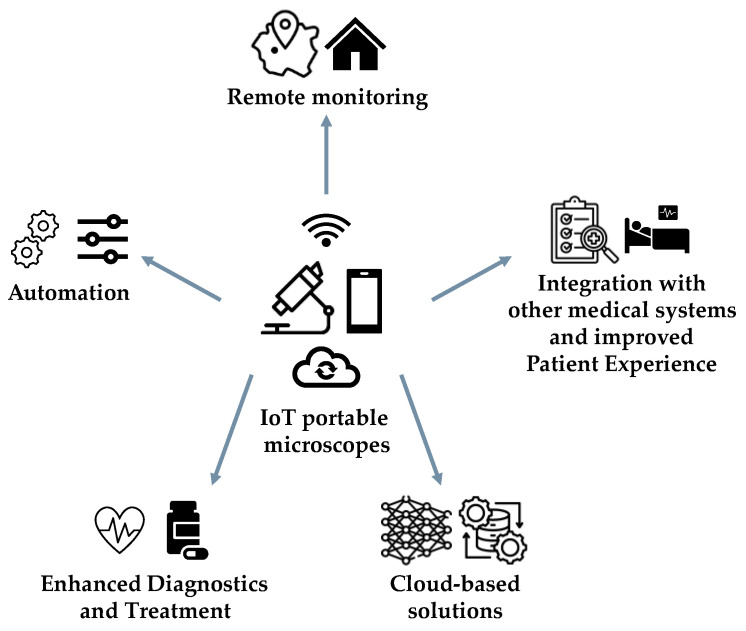
Applications and benefits of Internet-of-Things (IoT) integration with portable microscopy. IoT enables advanced features such as remote monitoring, integration with other medical systems and patient records, cloud-based data management and computing, and workflows automation. These innovations lead to enhanced diagnostics and treatment, overall improving patient experience.

**Figure 7 sensors-24-06682-f007:**
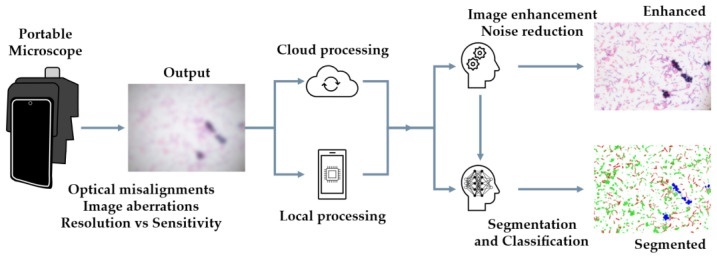
Image processing in portable microscopy. Key challenges include possible misalignments in the optical setup, image aberrations, and the trade-off between resolution and sensitivity, which degrade the output image quality. Image post-processing can be performed locally on the device or in the cloud, enhancing image quality, reducing noise, and enabling segmentation and classification tasks.

## Data Availability

No new data were created or analysed in this study. Data sharing is not applicable to this article.

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
