# Peer review of "Advances in Portable Optical Microscopy Using Cloud Technologies and Artificial Intelligence for Medical Applications"

_sensors, 2024, doi:10.3390/s24206682_

Round 1
Reviewer 1 Report
Comments and Suggestions for Authors
The Manuscript is devoted to rapidly growing field of gadget-like microscopes, which do not require experienced personnel both for image acquisition and for diagnostics, but instead, use machine learning algorithms and cloud technologies. The manuscript is not a research paper. It is such an ode to progress, which is important for advertising emerging technologies, and literature review, which is is useful for specialists interested in the field of portable and self-utilizing microscopy. I will not judge if the manuscript within the scope of this journal, but it is definitely interesting.
Nevertheless, the authors, probably being fascinated by the modern technologies, did not touch the first stage of any medical or biological investigation, namely, specimen preparation procedure. I do not require to describe the procedure itself, since it is definitely beyond the scope of the journal, but the authors should at least mention specimen preparation and caused limitations of usage of the technologies described in the manuscript.
Author Response
Dear Reviewer,
Thank you very much for taking the time to review this manuscript.
Please find the detailed responses below and the corresponding revisions and corrections in track changes in the re-submitted files.
Comments 1: The Manuscript is devoted to rapidly growing field of gadget-like microscopes, which do not require experienced personnel both for image acquisition and for diagnostics, but instead, use machine learning algorithms and cloud technologies. The manuscript is not a research paper. It is such an ode to progress, which is important for advertising emerging technologies, and literature review, which is is useful for specialists interested in the field of portable and self-utilizing microscopy. I will not judge if the manuscript within the scope of this journal, but it is definitely interesting.
Response 1: We appreciate the recognition of the relevance of our review in providing an overview of the advancements in portable microscopy and its applications for the specialists interested in the field. We have modified the Introduction [Page 2, Section 1, Lines 50-71] and Discussion [Page 21-22, Section 5, Lines 959-987] section to provide a more balanced view of portable solutions compared to conventional desktop microscopes, highlighting more of their limitations, including the issue of specimen preparation.
Comments 2: Nevertheless, the authors, probably being fascinated by the modern technologies, did not touch the first stage of any medical or biological investigation, namely, specimen preparation procedure. I do not require to describe the procedure itself, since it is definitely beyond the scope of the journal, but the authors should at least mention specimen preparation and caused limitations of usage of the technologies described in the manuscript.
Response 2: We thank the reviewer for the valuable suggestion on specimen preparation. We agree that it is an essential first step and it can indeed impose limitations on the applicability of portable microscopy. Therefore, we have included in the revised manuscript a paragraph in the Discussion section [Page 21, Section 5, Lines 959-970] to address how specimen preparation affects the use of these technologies.
Here is the revised text in the manuscript: “However, an important aspect that cannot be overlooked is specimen preparation. In medical and biological investigations, proper specimen handling is a critical pre-requisite for accurate imaging and diagnosis. Preparation techniques vary widely from sample to sample, with specific protocols and tools. While innovations in portable microscopy focus on simplifying image acquisition, visualization, analysis and interpretation, these instruments still require the sample to be manually prepared by experts, and the accuracy of the results depends on the correctness of the procedure [251]. This factor limits the ease of use in field settings and real-world applications, challenging the widespread adoption of portable microscopy. Despite some efforts to automate specimen preparation with microfluidic and optofluidic devices [251,252] or to simplify the sample preparation procedure for field acquisitions [253], this issue remains under-addressed in research on portable microscopy solutions.”
Reviewer 2 Report
Comments and Suggestions for Authors
1. The title should be corrected.
1.1. Since in addition to optical microscopes, there are also electronic scanning and transmission microscopes (SEM and TEM), which are used both in materials science and in medicine, the word "optical microscopy" should be added to the title.
1.2. «The Impact of Portability, IoT and AI» It is recommended to replace with "using cloud technologies and artificial intelligence"
The authors discuss the use of smartphones for microscopy mainly for medical purposes. The reviewer believes that this should be reflected in the title of the article.
2. Page 2. Lines 48-50 “ These include confinement to laboratory settings, high costs, dependence on skilled personnel ….
Reviewer: The text of this paragraph should be corrected. There are currently a large number of desktop optical microscopes (especially those made in China) that do not require a laboratory room and are so inexpensive that they can be purchased even for a home or school laboratory. Whereas a modern smartphone with a high-resolution camera costs much more than some desktop optical microscopes.
3. Page. 2, Lines 53-54 «They offer innovative solutions towards cost-effective and field-applicable devices».
Reviewer: In "field" conditions there may well be no Internet, in which case neither artificial intelligence nor cloud technologies will be available. The authors should be more objective and emphasize not only the disadvantages, but also the advantages of conventional desktop microscopes - such as their reliability, simplicity of design, independence from the presence of the Internet and the ability to work with digital cameras, when the image obtained in the microscope can be photographed on a digital matrix, then transferred to a computer and processed using modern computer image processing programs.
Conclusion.
The review, dedicated to the current topic of the transition from optical desktop to portable microscopes based on smartphone cameras, is done professionally and reveals the topic stated in the title.
Several comments indicated by the reviewer above require minor revision of the text.
The reviewer has only one doubt, one question - to what extent this review fits the theme of the journal issue "Sensors".
Author Response
Dear Reviewer,
Thank you very much for taking the time to review this manuscript.
Please find the detailed responses below and the corresponding revisions and corrections in track changes in the re-submitted files.
Comments 1: 1. The title should be corrected.
1.1. Since in addition to optical microscopes, there are also electronic scanning and transmission microscopes (SEM and TEM), which are used both in materials science and in medicine, the word "optical microscopy" should be added to the title.
1.2. «The Impact of Portability, IoT and AI» It is recommended to replace with "using cloud technologies and artificial intelligence"
The authors discuss the use of smartphones for microscopy mainly for medical purposes. The reviewer believes that this should be reflected in the title of the article.
Response 1: We appreciate the constructive feedback and have made the suggested modification to the title of the article, from “From Benchtop to Handheld Microscopy: The Impact of Portability, IoT and AI” to “Advances in Portable Optical Microscopy using Cloud Technologies and Artificial Intelligence for Medical Applications”.
Comments 2: 2. Page 2. Lines 48-50 “ These include confinement to laboratory settings, high costs, dependence on skilled personnel ….
Reviewer: The text of this paragraph should be corrected. There are currently a large number of desktop optical microscopes (especially those made in China) that do not require a laboratory room and are so inexpensive that they can be purchased even for a home or school laboratory. Whereas a modern smartphone with a high-resolution camera costs much more than some desktop optical microscopes.
Response 2: We thank the reviewer for this comment. We have modified the Introduction section to better reflect this observation. In the revised text, we have made a distinction between benchtop optical microscope solutions that have become more affordable, and those high-end, specialized and more expensive microscopes for which laboratory conditions and trained professionals are still required.
Here is the revised text in the manuscript [Page 2, Section 1, Lines 50-52]: “While affordable benchtop optical microscopes are increasingly available, there are also specialized high-end microscopes that present improved capabilities but come with higher costs and require skilled personnel for operation [15].”
Comments 3: 3. Page. 2, Lines 53-54 «They offer innovative solutions towards cost-effective and field-applicable devices».
Reviewer: In "field" conditions there may well be no Internet, in which case neither artificial intelligence nor cloud technologies will be available. The authors should be more objective and emphasize not only the disadvantages, but also the advantages of conventional desktop microscopes - such as their reliability, simplicity of design, independence from the presence of the Internet and the ability to work with digital cameras, when the image obtained in the microscope can be photographed on a digital matrix, then transferred to a computer and processed using modern computer image processing programs.
Response 3: We thank the reviewer for this comment. We have revised the paragraph to present a more balanced view of portable solutions integrated with IoT and AI, positioning them as complementary to laboratory-based devices [Page 2, Section 1, Lines from 56-71]. We have rephrased the paragraph indicating that compactness is a priority for field applications and that while desktop devices can operate independently of internet connectivity, integrated communication capabilities are a requirement for portable systems [Page 2, Section 1, Lines from 62-66]. At the end of the paragraph, we also highlight that while these innovations offer new opportunities, they present new challenges as well.
Here is the revised text in the manuscript: “Recent advancements in technology and evolving industry trends have introduced complementary solutions that are transforming the conventional chain of image acquisition and interpretation in microscopy, as illustrated in Figure 1. These innovations prioritize compactness and portability for field applications [17,18], with a special emphasis on the use of smartphones to enhance accessibility for both professional and non-expert users [19–21]. While tabletop microscopes have the advantage of operating independently of internet connectivity, relying instead on local connections to computers and digital cameras for image visualization and analysis, incorporated communication [16] and image processing [6] capabilities are trending solutions that are driving the development of portable microscopes. These systems integrate the Internet of Things (IoT) [18] to facilitate faster and more efficient data communication [22]. Coupled with cloud services [23] for data sharing and telemedicine [24] and deep learning (DL) algorithms for automated image processing and interpretation [25,26], these solutions offer new opportunities as well as new challenges for the field of microscopy.”
In addition, we have revised the Discussion section [Page 21-22, Section 5, Lines 959-987], acknowledging the limitations of portable microscopes including the problem of sample preparation and the dependency on internet connectivity. We have highlighted that, thanks to their advantages, desktop microscopes can still be the preferred solutions in these situations.
Here is the revised text in the manuscript: “However, an important aspect that cannot be overlooked is specimen preparation. In medical and biological investigations, proper specimen handling is a critical pre-requisite for accurate imaging and diagnosis. Preparation techniques vary widely from sample to sample, with specific protocols and tools. While innovations in portable microscopy focus on simplifying image acquisition, visualization, analysis and interpretation, these instruments still require the sample to be manually prepared by experts, and the accuracy of the results depends on the correctness of the procedure [251]. This factor limits the ease of use in field settings and real-world applications, challenging the widespread adoption of portable microscopy. Despite some efforts to automate specimen preparation with microfluidic and optofluidic devices [251,252] or to simplify the sample preparation procedure for field acquisitions [253], this issue remains under-addressed in research on portable microscopy solutions.
Several other challenges remain. One of these is data security and privacy. With the transmission and storage of sensitive medical data in the cloud, robust encryption and secure communication protocols are required. Additionally, relying on internet connection and cloud computing introduces issues related to network dependency with potential costs that can be prohibitive in resource-constrained settings. In such environments, this dependency potentially compromises the overall effectiveness of these systems by making cloud-based artificial intelligence and data transmission inaccessible. In contrast, traditional desktop microscopes can function entirely offline, and with their simpler design, they are often more robust and reliable, making them still the preferred solution where network access is poor or absent.
Improving machine learning and deep learning algorithms for image analysis is needed, not only in terms of accuracy but also in developing lightweight models that can be deployed on devices with limited hardware. Portable microscopes should be able to perform local computations and store the results temporarily for later transmission. Moreover, clinical validation is mandatory to ensure the accuracy of the system’s diagnostic capabilities, while continued research and development for cost-effective solutions is crucial to make this technology accessible to a wider audience.”.
Comments 4: Conclusion.
The review, dedicated to the current topic of the transition from optical desktop to portable microscopes based on smartphone cameras, is done professionally and reveals the topic stated in the title.
Several comments indicated by the reviewer above require minor revision of the text.
The reviewer has only one doubt, one question - to what extent this review fits the theme of the journal issue "Sensors".
Response 4: We thank the reviewer for the positive feedback. We have addressed the minor revisions as indicated above. Regarding your question about the relevance of this review to the journal’s theme, we believe that our review fits well within the scope, particularly in the areas of sensor-captured imaging, AI-enabled sensors and Internet of Things, as for the journal’s “Aims & Scope”. Portable microscopes, not just smartphone-based ones, rely heavily on advancements in sensor technology, enabling them to approach the capabilities of desktop microscopes. Our review covers the integration of portable microscopes with IoT and AI and these, at their core, are smart sensors that capture and transmit data in real time.